# Degree of Accuracy of the BMI Z-Score to Determine Excess Fat Mass Using DXA in Children and Adolescents

**DOI:** 10.3390/ijerph182212114

**Published:** 2021-11-18

**Authors:** David Monasor-Ortolá, Jose Antonio Quesada-Rico, Ana Pilar Nso-Roca, Mercedes Rizo-Baeza, Ernesto Cortés-Castell, Asier Martínez-Segura, Francisco Sánchez-Ferrer

**Affiliations:** 1Department of Nursing, University of Alicante, 03690 Alicante, Spain; david.monasor@ua.es (D.M.-O.); rizo.mercedes@gmail.com (M.R.-B.); asiertijo@gmail.com (A.M.-S.); 2Department of Clinical Medicine, Miguel Hernández University of Elche, 03550 Elche, Spain; jquesada@umh.es; 3Department of Pharmacology, Pediatrics and Organic Chemistry, Miguel Hernández University of Elche, 03550 Elche, Spain; ernesto.cortes@umh.es (E.C.-C.); f.sanchez@umh.es (F.S.-F.); 4Department of Pediatrics, General University Hospital San Juan de Alicante, 03550 Alicante, Spain

**Keywords:** BMI Z-score, childhood obesity, body composition, fat mass, growth charts

## Abstract

Obesity is caused by fat accumulation. BMI Z-score is used to classify the different degrees of weight status in children and adolescents. However, this parameter does not always express the true percentage of body fat. Our objective was to determine the degree of agreement between the fat mass percentage measured by DXA and the stratification of weight according to BMI Z-score in the pediatric age group. We designed a descriptive cross-sectional study. The patients were classified as underweight/normal weight with Z-scores between −2 and +0.99, overweight from 1 to 1.99, obese from 2 to 2.99, and very obese ≥3. We included 551 patients (47% girls), with a mean age of 11.5 ± 2.8 years (3.7–18 years). Higher BMI Z-scores were associated with a higher percentage of total fat (*p* < 0.001). However, there were important overlaps between both parameters, such that the BMI Z-score classified patients with the same percentage of total fat mass as having a different nutritional status classification. In conclusion, the stratification of weight status according to BMI Z-score revealed that 46.7% of patients had a fat percentage that did not correspond to their classification. For a more accurate weight assessment in clinical practice, we recommend combining anthropometric indices with diagnostic tools that better correlate with DXA, such as electrical bioimpedance.

## 1. Introduction

Childhood obesity is one of the most serious public health problems of the 21st century, and among non-communicable diseases it is a major concern [1]. The World Health Organization (WHO) Commission on Ending Childhood Obesity estimated in 2017 that 41 million children were obese or overweight. This figure is on the rise and is expected to climb to 70 million by 2025 [2].

Childhood obesity, defined as an excessive accumulation of energy in the form of fat in the pediatric population, is associated with a wide range of serious health complications and an increased risk of early onset of diseases such as diabetes, dyslipidemia, and cardiovascular disease, among others [1,3,4].

Obesity is a preventable disease and policies, environments, schools, and communities are fundamental for adequate health promotion. The detection of overweight and obese children should be a priority for health systems to prevent the associated complications [1]. The identification of obesity and the analysis of body composition by means of validated tools reduces the variability of clinical practice and establishes a common approach to prevention, diagnosis, treatment, and health promotion among the different groups of professionals who treat this condition [5].

The analysis of body composition is, therefore, fundamental to assess nutritional status and detect risk situations in children and adolescents. However, methods for direct measurement of the fat compartment are complex and are not commonly used in routine clinical practice. The gold standard for direct assessment of body composition in a safe way, with very low radiation and a low margin of error (3–6%), is dual X-ray absorptiometry (DXA) [6,7,8]. This tool estimates body composition (percentages of total fat, trunk fat, android fat, and gynoid fat) as well as determining the bone mineral density of the body [8].

Given its complexity, it is not frequently used in routine clinical practice. Instead, obesity is usually assessed by indirect indicators of body fat. The most widely used method for diagnosis in children and adolescents, due to its simplicity and low cost, is the body mass index (BMI) and its Z-score adjusted for age and sex, as established by the WHO, classified according to the percentile in which the patient falls [9,10].

However, some studies state that the BMI Z-score has low sensitivity for the detection of excess adiposity in children [11,12,13,14,15]. It may not be an ideal tool in some situations involving altered body composition [12]. In the case of children with Down syndrome, for example [13], whose muscle mass and bone mineral density are lower, the BMI Z-score provides underestimated values of obesity, without correlating with the data obtained by DXA. Likewise, it also appears to underestimate overweight and obesity status in children with infantile cerebral palsy, who have lower fat-free mass compared to typically developing children [14]. Furthermore, the sensitivity of the BMI Z-score for the identification of malnutrition status is low (47.4%) [15].

In addition, BMI data have been applied empirically, which can lead to confusion in the classification of patients. Moreover, the same BMI percentile could be different according to age, sex or race-ethnic groups [16,17]. Thus, children who are not obese may be classified as obese and vice versa, with the risk that a percentage of the pediatric population is not being included in health programs for childhood obesity [11,12,13,14,15]. However, there are few previous studies that evaluate the precision of the BMI z-score in terms of predicting the percentage of body fat in children. We, therefore, consider it important to clarify an adequate stratification in the pediatric population.

Accordingly, our main objective was to determine the degree of agreement between the fat mass percentage estimated by DXA and weight stratification according to the BMI Z-score in the pediatric age group. We also aimed to analyze the correlation between the BMI Z-score and the percentage of total fat mass and to determine the accuracy of the BMI Z-score to detect true excess fat mass according to DXA.

## 2. Materials and Methods

### 2.1. Study Design, Population and Sample

This was a descriptive cross-sectional study including patients between 3 and 18 years of age treated at the Growth and Metabolism Unit of San Juan de Alicante University Hospital, between January 2013 and December 2020. This unit is a reference in its health area and has a coverage of about 35,000 pediatric patients [18]. Inclusion criteria were as follows: having undergone a body composition measurement by DXA and the availability of the study variables in the corresponding clinical history.

### 2.2. Procedure

Demographic and anthropometric variables were collected including age, sex, weight, and height. Data on the level of pubertal maturation or exercise were not available. Total body fat percentage was also obtained. Weight was recorded without clothing or shoes with a scale DC-430 (Tanita^®^: Middlesex, UK) (error +/−0.1 kg) and height with a stadiometer 213, (Seca^®^: Birmingham, UK) (error +/−0.5 cm). BMI was calculated from the anthropometric data, and the BMI Z-score was calculated using the Seinaptracker^®^ program (Medicalsoft Intercath SL, University of Barcelona, Spain 2007–2008), using the WHO growth charts as a reference [19]. Estimation of body composition was offered to all patients treated for obesity in the consultation and it was carried out to those who accepted voluntarily. It was performed with the General Electric Lunar DXA densitometer, model DPXN PROTM (GE Healthcare, Little Chalfont, Buckinghamshire, UK). Based on the BMI Z-score, patients were classified as: underweight/normal weight with Z-scores between −2 and +0.99, overweight from 1 to 1.99, obese from 2 to 2.99, and very obese ≥ 3 [19,20].

### 2.3. Statistical Analysis

The mean fat percentage was calculated together with its 95% confidence interval at each BMI Z-score level (thin-normal/overweight/obese/very obese), applying the Kruskal–Wallis test to evaluate differences in mean values, due to lack of normality of total fat. A cubic polynomial model was fitted to explain the total fat percentage from the BMI Z-score, showing the R2 and the coefficients of the model. To obtain the Z-score classification cut-off points from the total fat percentage, ordinal multinomial models were fitted, showing the overall classification rate, together with its 95% confidence interval. The figures corresponding to each analysis are shown. Data analysis will be performed using R software v.4.0.2 (Center for Statistics, Frederiksberg, Denmark). For the modeling of the polynomial model the *stats* library was used, and for the multinomial ordinal model the *VGAM* library.

### 2.4. Ethical Issues

The study is included in the research line “body composition in pediatric ages by DXA”, approved by the Ethics Committee of the Hospital Universitario San Juan de Alicante with reference number 16/305. The data obtained from the usual clinical follow-up protocols for these patients were used in the study; therefore, the committee did not consider it necessary to obtain the express informed consent of the legal guardians. The criteria of the Declaration of Helsinki were followed at all times and, in accordance with the regulations, the data were anonymized and untraceable.

## 3. Results

Of 553 patients who had undergone DXA, a total of 551 (99.6%) had all the variables analyzed available and were included in the study: 259 girls (47%) and 292 boys (53%), with a mean age of 11.5 ± 2.8 years (3.7–18 years). According to the BMI Z-score, 53 patients were classified as underweight/normal weight (9.6%), 77 as overweight (14.0%), 184 as obese (33.4%), and 237 as very obese (43%). Higher BMI Z-scores were associated with a higher percentage of total fat. The Kruskal–Wallis test indicated that these differences were significant (*p* < 0.001) (Figure 1).

To test this association, we performed a cubic regression analysis of fat percentages in relation to BMI Z-score (Figure 2). The BMI Z-score explained 54.8% of the variability of total fat (R2 = 0.548) in the cubic fit, with a good goodness of fit of the model (*p* < 0.001), showing stabilization around 50% of total fat, starting from a BMI Z-score of approximately 4.

However, significant overlaps were found between fat percentage and the groups classified according to BMI Z-score. This can be seen in Figure 3. The probability plot reflects the low degree of sensitivity of these cut-off points. Thus, the BMI Z-score classifies patients with the same percentage of total fat mass as being in different nutritional status categories.

To detect the cut-off point for the fat percentage that truly differentiates each weight group by BMI Z-score, a fit analysis was performed using ordinal multinomial models. The optimal cut-off points found are expressed in Table 1. Thus, in our study, the equivalence would be as follows: a BMI Z-score < 1 corresponds to a total fat mass of less than 25.5%, between 1 and 1.99 is equivalent to a fat mass of 25.5–33%, 2–2.99 to a fat mass of 33–43.5%, and a BMI Z-score ≥ 3 corresponds to a total fat mass greater than 43.5%.

## 4. Discussion

### 4.1. Summary

The mean body fat percentages were statistically different for the groups classified according to BMI Z-score in children and adolescents. Furthermore, an increasing and curvilinear association was observed between total fat percentage and BMI Z-score, stabilizing at 50% total fat starting from a BMI Z-score value of 4.

Using ordinal multinomial models, we propose total fat percentage cut-off points of 25.5, 33.0, and 43.5 for overweight, obese, and very obese, respectively, classified by BMI Z-score, with an overall accuracy rate between the two classifications of 53.4%. It is likely that some pediatric patients are misclassified as obese by BMI Z-score, with respect to the actual percentage of total fat.

### 4.2. Strengths and Limitations

The strengths of our study include the use of DXA as a technique for measuring body fat percentage, the large number of cases studied in a wide range of pediatric ages, and the statistical methodology used. The main limitation of our study is selection bias, as we obtained a sample of the population that is not representative of the population due to the way in which it was obtained in a pediatric practice that treats pediatric patients with nutritional problems, especially overweight and obesity. Similarly, as this was a cross-sectional study, we do not have data on the evolution of adiposity in the patients with age or with treatment, although this was not one of the objectives of the study. Finally, we do not have specific data on the stage of pubertal maturation or on the level of exercise. This means that we cannot assess how these variables influence the distribution of body fat.

### 4.3. Comparison with the Literature

Methods of direct measurement of the fat compartment, such as DXA, are impractical in routine clinical management. For this reason, simpler anthropometric parameters are generally used, such as the BMI Z-score, the diagnostic accuracy of which is still questioned, especially in the pediatric population. Given the increase in childhood obesity and its associated complications in adulthood, early detection is of great importance for early intervention, which is why it is important to use precise diagnostic criteria agreed upon by scientific societies [21,22,23]. This is especially true when the presence of symptoms associated with the metabolic syndrome begin to be present during childhood in a significant number of children with obesity [4].

We consider the present study of great interest because it evaluates the suitability of the BMI Z-score for detecting excess adiposity in children and, consequently, its accuracy as a clinical diagnostic tool. In specialized care, a complete study of the direct measurement of the body composition of the child is essential to evaluate nutritional status and also to detect risk situations. The gold standard for this is DXA, which provides data on the percentage of total fat, trunk fat, android fat, and gynoid fat, in addition to determining the bone mineral density of the whole body [6,7,8]. The use of electrical bioimpedance, which appears to correlate well with DXA [24] or magnetic resonance imaging [25], is also considered valid.

Few studies have established clear cut-off points for defining obesity by DXA in children and adolescents. Tello-Winniczuk et al. [26] and Velázquez-Alba et al. [27] propose diagnosing obesity starting from a total fat percentage of 35%; however, these studies were carried out in adults. William et al. [3] in 1992 had already shown that a body fat percentage higher than 25% in boys and 30% in girls was a risk factor for cardiovascular disease (*p* < 0.05), without defining these patients as obese. The work of Ryder et al. [28], which was performed in children and adolescents, established a median 95th BMI percentile of approximately 39% total fat in girls and 32% in boys, when referenced through lines in percentile charts. Meanwhile, Cossio-Bolaños et al. [29], provided lower values for the 95th BMI percentile of 29.7% total fat (segregated by sex in girls 35.2% and boys 26.1%). In our study, according to the BMI Z-score, the patient is classified as obese with a body fat percentage of 33% or higher, an intermediate percentage to those of the studies analyzed. Taking these parameters into account, our study found that the classification of weight status by BMI Z-score leads to erroneous classification of children with respect to these fat percentages.

There are also several clinical conditions or disorders in which it has been shown that the correlation between DXA and BMI Z-score is not satisfactory, such as in patients with Down syndrome [13] in whom the BMI underestimates the alteration of the fat percentage in up to 57% of cases. Likewise, the study by Duran et al. [15], in children with cerebral palsy, showed that the BMI Z-score overestimated the prevalence of underweight. In addition, a study on the accuracy of the BMI Z-score for detecting changes in fat percentage after a therapeutic intervention showed that this parameter had low specificity and, therefore, was not a solid predictor of adiposity change [12].

This lack of accuracy of the BMI Z-score was already acknowledged in patients with altered body composition due to their baseline condition as described above. However, in children without these alterations, it has continued to be used as the main anthropometric parameter. Our study also highlights the limitations of this marker for estimating body composition in children and adolescents with obesity.

Diagnostic accuracy in childhood obesity is a challenge. Our data demonstrate the limitation of the BMI Z-score to stratify the degree of adiposity in patients. Moreover, given that the WHO growth charts are based on the BMI Z-score to define overweight and obesity, our contribution is essential as it provides data on the cut-off point for adiposity to which the BMI standard deviations correspond. Although the growth charts established by the WHO provide guidance on the characteristics of population growth, the ideal is to use the standards for our own reference population [30], hence the importance of studies such as ours.

### 4.4. Implications for Clinical Practice and Research

Diagnostic accuracy in childhood obesity is fundamental. Studies, such as ours, that evaluate the accuracy in relation to body fat percentage of a parameter as widely used as the BMI Z-score, are essential. The results obtained indicate the need for an analysis of fat percentage as a complement to the BMI Z-score parameter in all patients suspected of being overweight and with different degrees of obesity. Much more accessible techniques such as bioimpedance could be used for this measurement.

We, therefore, consider it of great interest to determine the total fat percentage values that allow a better classification of overweight in the pediatric age group. The implication for clinical practice is crucial since a precise classification of the obese patient is essential to determine the risk of complications and the specific therapeutic approach. This would allow us to optimize the diagnosis and monitoring of pediatric patients with nutritional problems.

## 5. Conclusions

The stratification of weight status according to BMI Z-score in our sample had an accuracy rate of just 53.4%, showing a high number of patients with the same fat percentage classified differently according to BMI Z-score. For a more accurate weight assessment in clinical practice, we recommend combining anthropometric indices with diagnostic tools that correlate more closely with DXA, such as electrical bioimpedance.

## Figures and Tables

**Figure 1 ijerph-18-12114-f001:**
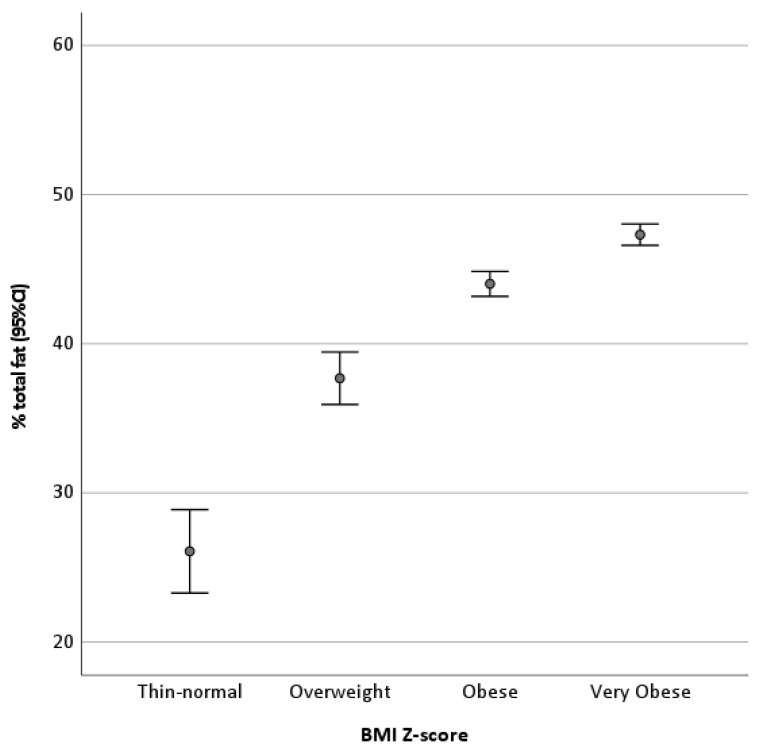
Percentage of body fat (mean ± 95%CI) according to nutritional status classification by BMI Z-score.

**Figure 2 ijerph-18-12114-f002:**
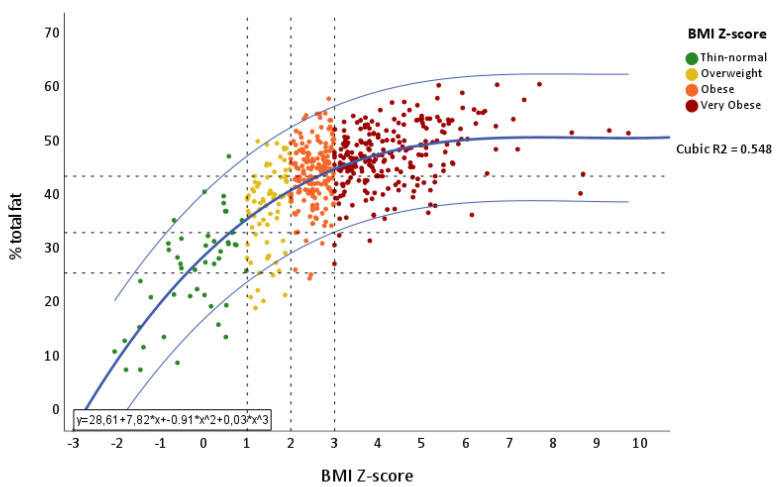
Curvilinear adjustment of BMI Z-score over total fat percentage, showing the cut-off points according to both cut-off points.

**Figure 3 ijerph-18-12114-f003:**
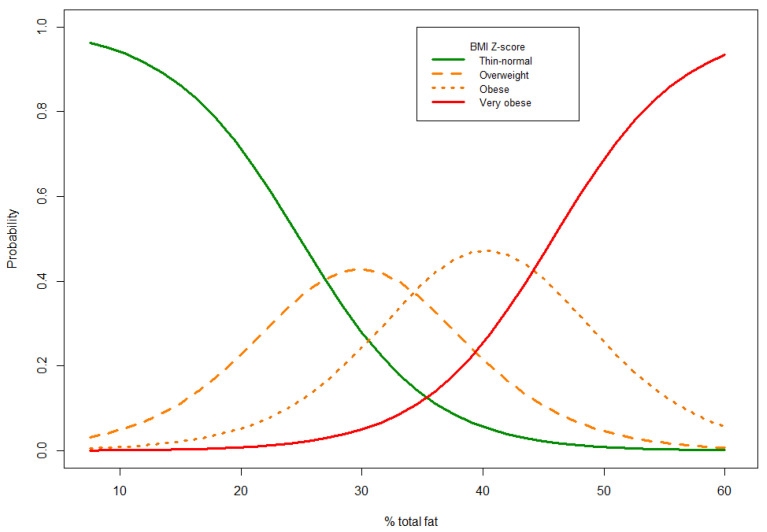
Probability plot for classification into the different BMI Z-score groups according to total fat percentage.

**Table 1 ijerph-18-12114-t001:** Optimal cut-off points for fat percentage that differentiate weight groups according to BMI Z-score.

Weight Group According to BMI Z-Score (SD)	Total Fat	
	Cut-Off (%)	Probability of Accuracy
Overweight (1–1.99)	25.5	47.1
Obese (2–2.99)	33.0	39.9
Very obese (≥3)	43.5	44.0
Percentage of accuracy (95%CI)	53.4 (49.2–57.6)	

## Data Availability

The data presented in this study are available on request from the corresponding author. The data are not publicly available due to privacy.

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
