# Peer review of "Degree of Accuracy of the BMI Z-Score to Determine Excess Fat Mass Using DXA in Children and Adolescents"

_ijerph, 2021, doi:10.3390/ijerph182212114_

Round 1
Reviewer 1 Report
dear authors.
this is a very interesting study - that offers very few novel findings to the literature.
from my knowledge of other studies, the value and limits of bmiz in children and adolescents are well understood as a result of these studies. while bmiz is sensitive to changes in adiposity, its poor specificity makes it a poor predictor of changes in total body fat (percent fat). as a result, physicians must be cautious when monitoring changes in a developing child's body composition with bmiz alone to avoid misclassifying or overlooking significant changes.
however, the paper is well written and some changes needed first before it suitable for release.
title need to be changed. bmiz is not accurate. 50% is not accurate. this should be upfront as statement in the title.
abstract: acceptable
keywords: maybe consider adding growth charts, obesity, z-scores.
introduction: bmi-for-age z-scores (bmiz) have been widely used in all types of analyses, including obesity interventions - this should be discussed critically in the results.
bmi transformations should be clearly described in the methods.
provide reference for very obese category.
the statistical analysis section needs to be rewritten for clarity. modelling need to be described in details. information about used software and also any codes will be needed.
e.g. kruskall wallis test was used without details on normality check. i would have preferred if anova/ancova is performed.
after line 224 you authors need to have a paragraph on the clinical implications of this to healthcare. this will be major point to make this paper contributory to literature.
conclusion is adequate
Author Response
Dear Reviewer:
Below, we describe the responses to the reviewers’ comments and the modifications made to our manuscript ijerph-1414156, entitled “Accuracy of the BMI Z-score to determine excess fat mass using DXA in children and adolescents” which we believe have considerably improved the work:
Reviewer 1:
This is a very interesting study - that offers very few novel findings to the literature. The paper is well written and some changes needed first before it suitable for release:
- Title need to be changed. bmiz is not accurate. 50% is not accurate. this should be upfront as statement in the title. We have modified the title to better understand the intention to assess the degree of precision of the BMI.
- Keywords: maybe consider adding growth charts, obesity, z-scores. In the keywords section, we have specified "childhood obesity", "BMI z-score" and we have added "growth charts".
- Introduction: bmi-for-age z-scores (bmiz) have been widely used in all types of analyses, including obesity interventions - this should be discussed critically in the results. Indeed, it is a point to explain. It is detailed on lines 205-212 of the discussion.
- BMI transformations should be clearly described in the methods. The transformation of the BMI is detailed in lines 94-96 of the methods section.
- Provide reference for very obese category. We have added bibliographic reference referring to the group of "very obese" [19-20].
- The statistical analysis section needs to be rewritten for clarity. modelling need to be described in details. information about used software and also any codes will be needed. g. kruskall wallis test was used without details on normality check. i would have preferred if anova/ancova is performed. The percentage of total fat presents asymmetry and a lack of normality adjustment, for this reason a non-parametric method has been applied. This asymmetry is observed in figure 2. A mention of the lack of normality has been added in the Methods section. The statistical program R used and the libraries for modeling have been added.
- After line 224 you authors need to have a paragraph on the clinical implications of this to healthcare. this will be major point to make this paper contributory to literature. We have added an explanation of this point in lines 237-239.
Reviewer 2 Report
This is an interesting study that examines how accurately the Z-score of body mass index (BMI) predicts total body fat using dual X-ray absorptiometry (DXA) as the golden standard.
However, there are several major problems with the paper that should be corrected.
1. In the introduction, the authors state that various problems with the BMI Z-score have already been pointed out in previous studies.If that is the case, why was this study necessary?
The authors should clearly state what has not been clarified in previous studies and what they wanted to clarify in this study.
2. As the author has already mentioned, there are several major limitations in this study.
One is that the target population was not healthy children, but a special population that was treated at a children's hospital and underwent DXA.
Was there a protocol in place to determine whether or not to perform DXA?
The authors should consider which pediatric patients in this study were most likely to receive DXA and how this would affect the results and include it in Limitation.
3. Was there a uniform protocol for weight measurement within the study period? For example, were the measurements done in clothes?
4. There is also a question about the statistical analysis.
Since the BMI Z-score is a simple index, it is better to seek sensitivity and specificity rather than accuracy.
In other words, the BMI Z-score is a screening test that should be used as an indicator of whether or not to perform more detailed tests such as DXA.
I believe that it would be useful in clinical practice to calculate which criteria should be used to recommend a detailed examination such as DXA for suspected obesity (sensitivity), and which criteria should be used to determine if a detailed examination is unnecessary (specificity).
Author Response
Dear Reviewer:
Below, we describe the responses to the reviewers’ comments and the modifications made to our manuscript ijerph-1414156, entitled “Accuracy of the BMI Z-score to determine excess fat mass using DXA in children and adolescents” which we believe have considerably improved the work:
Reviewer 2
This is an interesting study that examines how accurately the Z-score of body mass index (BMI) predicts total body fat using dual X-ray absorptiometry (DXA) as the golden standard. However, there are several problems with the paper that should be corrected.
- In the introduction, the authors state that various problems with the BMI Z-score have already been pointed out in previous studies. If that is the case, why was this study necessary? The authors should clearly state what has not been clarified in previous studies and what they wanted to clarify in this study. We have added a sentence in the introduction explaining our novel contribution in the approach to the topic.
- As the author has already mentioned, there are several limitations in this study. One is that the target population was not healthy children, but a special population that was treated at a children's hospital and underwent DXA. Was there a protocol in place to determine whether or not to perform DXA? The authors should consider which pediatric patients in this study were most likely to receive DXA and how this would affect the results and include it in Limitation. We have specified this in the "procedure" section.
- Was there a uniform protocol for weight measurement within the study period? For example, were the measurements done in clothes? All patients were weighed without clothing or shoes on the same calibrated precision scale. We have added this information in the methods section.
- There is also a question about the statistical analysis. Since the BMI Z-score is a simple index, it is better to seek sensitivity and specificity rather than accuracy. In other words, the BMI Z-score is a screening test that should be used as an indicator of whether or not to perform more detailed tests such as DXA. I believe that it would be useful in clinical practice to calculate which criteria should be used to recommend a detailed examination such as DXA for suspected obesity (sensitivity), and which criteria should be used to determine if a detailed examination is unnecessary (specificity). We agree with this comment but we had to use accuracy because the model is multinomial, not binary. In the case of multinomial classification, it is not possible to calculate parameters such as sensitivity or specificity, and for this reason accuray has been calculated.
Reviewer 3 Report
Abstract
Please avoid breaking syllables of words at the end of lines. For example: Line 23 and line 24
Review it throughout the text of the article.
In general the abstract reports well on what the research is about and the main findings found
Introduction
Line 59-60. Indicates that there are several studies in which the BMI Z-score has a low sensitivity, but only includes citation number 11. This should be expanded to these "several studies".
Line 67. Remove the full stop after the parenthesis
Lines 68-69. Should better explain the meaning of this statement and include numerous references which is what the meaning of the sentence indicates and not just quote number 16.
Materials and Methods
This section gives a good indication of how the research was carried out and has the relevant ethical references.
However, the characteristics of the sample do not indicate, on the one hand, the degree of sexual maturity they may have (Tanner scale), which may alter the body percentages, as well as the type of physical exercise that alters muscle mass or sedentary habits.
These variables have not been controlled and therefore there is an evident bias that has not been taken into account.
Results, Discussion and Conclusion
This section is supported by the methodology and the findings are probably framed as false true as the maturity and exercise variables were not controlled for.
Author Response
Dear Reviewer:
Below, we describe the responses to the reviewers’ comments and the modifications made to our manuscript ijerph-1414156, entitled “Accuracy of the BMI Z-score to determine excess fat mass using DXA in children and adolescents” which we believe have considerably improved the work:
Reviewer 3
- Please avoid breaking syllables of words at the end of lines. For example: Line 23 and line 24. Review it throughout the text of the article. We have removed the hyphens at the end of the sentence to avoid breaking syllables.
- Line 59-60. Indicates that there are several studies in which the BMI Z-score has a low sensitivity, but only includes citation number 11. This should be expanded to these "several studies". We have added more citations to this sentence.
- Line 67. Remove the full stop after the parenthesis. We have removed it.
- Lines 68-69. Should better explain the meaning of this statement and include numerous references which is what the meaning of the sentence indicates and not just quote number 16. We have better explained the phrase and we have added a bibliographic citation.
- Materials and Methods. The characteristics of the sample do not indicate, on the one hand, the degree of sexual maturity they may have (Tanner scale), which may alter the body percentages, as well as the type of physical exercise that alters muscle mass or sedentary habits. These variables have not been controlled and therefore there is an evident bias that has not been taken into account. We have added comments about this issue in the methods section.
- Results, Discussion and Conclusion. This section is supported by the methodology and the findings are probably framed as false true as the maturity and exercise variables were not controlled for. We have added comments about this issue in the discussion.
Round 2
Reviewer 2 Report
The authors have revised the paper well. I have no further comments to add.
Reviewer 3 Report
Thank you very much for considering my suggestions.
Perhaps you should give a somewhat more extensive and substantiated explanation with some references to not having taken into account either maturation or exercise practice.